# Loss of Blink Regularity and Its Impact on Ocular Surface Exposure

**DOI:** 10.3390/diagnostics13142362

**Published:** 2023-07-13

**Authors:** Genis Cardona, Marc Argilés, Elisabet Pérez-Cabré

**Affiliations:** 1Optics and Optometry Department, Universitat Politècnica de Catalunya, c/Violinista Vellsolà, 37, 08222 Terrassa, Spain; marc.argiles@upc.edu (M.A.); elisabet.perez@upc.edu (E.P.-C.); 2Applied Optics and Image Processing Group (GOAPI), Universitat Politècnica de Catalunya (UPC), 08222 Terrassa, Spain; 3Center for Sensors, Instruments, and Systems Development (CD6), Universitat Politècnica de Catalunya (UPC), 08222 Terrassa, Spain

**Keywords:** blink regularity, ocular surface exposure, spontaneous eyeblink rate, tear film break-up time

## Abstract

(1) Background: Changes in blink parameters have been found to influence ocular surface exposure, eliciting symptoms of dry eye and ocular signs. The aim of the study was to highlight the relevance of including blink regularity as a new parameter to fully characterize blinking; (2) Methods: A novel characterization of blink parameters is described, including spontaneous eyeblink rate (SEBR), percentage of incomplete blinks, and blink regularity. A pilot study was designed in which tear film break-up time (TFBUT), blink parameters, and the time percentage of ocular surface exposure were determined in eight subjects (52.0 ± 16.6 years, 4 females) in three experimental conditions (baseline, reading a hard-copy text, and reading from an electronic display). Blink parameters were monitored through asynchronous image analysis of one minute video segments; (3) Results: All blink parameters were influenced by experimental conditions. A trend was observed in which both reading tasks resulted in an increase in ocular surface exposure, mainly related to a combination of reduced SEBR, increased percentage of incomplete blinks, and loss of blink regularity; (4) Conclusions: A complete characterization of blink parameters is relevant to better understand ocular dryness related to surface exposure and to advice patients towards a reeducation of their blinking habits.

## 1. Introduction

The Ocular Protection Index (OPI) was developed by Ousler and co-workers in 2002 as a tool to assess the risk of ocular surface damage due to exposure [1,2]. As such, the OPI is determined by dividing the tear film break-up time (TFBUT) by the mean interblink interval (<IBI>) (Equation (1)), whereupon any value (dimensionless) equal or superior to 1 denotes a healthy ocular surface, as blinking and tear film renewal occur before tear film break-up and ocular surface exposure.

OPI = TFBUT/<IBI>.(1)

Subsequently, the OPI has shown good accuracy and reproducibility to evaluate differences between tear film substitutes [3,4], as well as to assess ocular surface stability, and its relationship with visual asthenopia, while viewing 3D displays [5].

In 2011, some of the authors involved in the development of the OPI described a series of limitations of the traditional technique which led to the conception of the Ocular Protection Index 2.0 System [6,7]. Most notably, in the original OPI, the mean IBI was determined by considering all eyeblinks as complete, and a single non-synchronous measurement of tear film break-up time was introduced in the numerator. Conversely, the OPI 2.0 uses a digital camera to capture 1 min video segments and subsequent image analysis to identify both complete and incomplete eyeblinks and to determine the fraction of the corneal surface showing evidence of tear film break-up at the end of each individual IBI. The OPI 2.0 value for each 1 min video segment (in %/s) is calculated by dividing the average of the percentage of the exposed cornea by the corresponding mean IBI value (Equation (2)).
OPI 2.0 = <% Exposed ocular surface>/<IBI>.(2)

With the OPI 2.0 System, the authors reported higher values for dry eye patients than normal subjects [7]. However, the calculation of the OPI 2.0 involves a higher computational effort than that of the original OPI since it is necessary to evaluate the area of corneal surface exposure with time after tear-film breakup. Moreover, OPI 2.0 measurements are conducted while patients sit behind a slit-lamp incorporating a cobalt blue filter and require sodium fluorescein instillation, that is, results may not be extrapolated to real-life conditions.

It must be noted that the developers of both the OPI and the OPI 2.0 System recommend measuring the IBI while subjects observe an Early Treatment Diabetic Retinopathy Study (ETDRS) chart or perform a standard visual task such as watching a documentary. However, the choice of visual task is not inconsequential. Indeed, spontaneous eyeblink rate (SEBR) has been shown to be highly sensitive to cognitive and ambient conditions, with authors reporting increased values during conversation, in anxiety states, and in dry eye patients [8,9]. On the contrary, cognitively demanding tasks such as reading and using visual display terminals have been documented to lower SEBR values [9,10,11,12,13]. Bentivoglio and co-workers, for instance, reported changes in SEBR from 26 blinks/minute during conversation to 17 blinks/minute in silent primary gaze, further decreasing to 4.5 blinks/minute while reading [14]. Doughty described significant differences in eyeblink regularity between the previous three tasks, with eyeblinks during conversation displaying a highly irregular behavior in which most eyeblinks were grouped into short sequences [9]. Interestingly, Nakano and coworkers explored eyeblink patterns while participants watched a video and suggested the presence of an internal mechanism to regulate the most appropriate time for a blink in order to minimize the chance of losing critical information given a continuous stream of visual data [15].

In addition to SEBR and eyeblink regularity, eyeblink amplitude is gaining relevance, as recent studies suggest that incomplete blinking is closely related to the use of electronic devices [16,17,18,19]. For instance, Argilés and co-workers described a higher incidence of incomplete blinking when subjects used a computer display or a tablet to read than when subjects read the same text in hard-copy format [17]. Also, Hirota et al. reported an association between incomplete blinking and tear film instability in computer users [19]. These studies suggest that incomplete blinking, rather than decreased SEBR, may be the main causative factor of visual fatigue and dry eye in visual display terminal operators.

Therefore, the joint evaluation of SEBR, blink amplitude and blink regularity may be essential to advance the understanding of the dry eye symptomatology associated with, amongst other conditions, computer use. As such, the OPI and OPI 2.0 approaches may yield insufficient information to characterize these blink parameters: given the same mean IBI, for example, high OPI 2.0 values may be interpreted either as high irregular blinking or as a large percentage of incomplete eyeblinks, both resulting in larger areas of ocular surface exposure. This indetermination may hinder any real effort at managing dry eye patients via re-education of blinking behavior. It was the aim of the present study to develop the proof of concept of a complete characterization of ocular surface exposure in the context of three different blink parameters: SEBR, blink amplitude, and blink regularity. Consequently, a small pilot study was conducted to assess the differences in these parameters and their impact on ocular surface exposure in three different experimental conditions: looking straight-ahead in silence, reading a text presented on a computer display and reading the same text in hard-copy format.

## 2. Materials and Methods

### 2.1. Blink Parameters: Spontaneous Eyeblink Rate, Amplitude and Regularity

It is the aim of this section to describe the impact on ocular surface exposure of SEBR, blink amplitude, and blink regularity, as well as TFBUT. Figure 1 displays four examples of different blink patterns, within a 1 min segment timeframe (SEBR, mean IBI, and TFBUT are kept constant, at 6 blinks per minute, 12 s and 10 s, respectively). To simplify the description, in these examples the first and last blinks are set to occur at the 0 s and 60 s marks, respectively, that is, exactly at the beginning and end of each 1 min segment (please note that in a real recording blinks rarely will be synchronous with the start and end of a video segment, as will be discussed below). Complete blinks (in which none of the cornea is visible on blink completion) [18] are shown as continuous blue lines, whereas incomplete blinks are represented by discontinuous red lines. The examples show different patterns of blink regularity, defined as the coefficient of variation of IBI (i.e., the standard deviation of IBI divided by its mean: values under 100% may be considered to denote low variance or moderate to high regularity, whereas values over 100% correspond to low regularity, with a wide dispersion of eyeblinks within the explored time-frame).

In example A, blinking is very regular, with a blink every 12 s, and all blinks are complete. Given a TFBUT of 10 s and individual IBI values of 12 s, ocular surface exposure will occur during 2 s before each blink (shown in the figure as a shadowed area), or during 10 s over the total of 60 s (10/60 or 16.7% of the time). In example B, although a blink event also occurs every 12 s, some of the blinks are incomplete, thus failing to lead to tear film renewal and redistribution. Thus, considering complete blinks only, IBI values are 12, 36, and 12 s, and ocular exposure time is 2, 26, and 2 s (shadowed areas), or 30 s over a total of 60 s (30/60 or 50% of the time). Examples C and D show very irregular blink patterns, with clusters of blinks and periods without blinking. In example C, individual IBI values are 6, 3, 42, 3, and 6 s, resulting in a total ocular exposure time of 32 s (32/60 or 53.3% of the time). Finally, in example D, incomplete blinking is added to irregular blinking, with IBI values when considering only complete blinks of 6, 48, and 6 s and total ocular exposure of 38 s over 1 min (38/60 or 63.3% of the time).

Table 1 summarizes the blink and tear film parameters of the four examples under consideration. It may be observed that TFBUT, SEBR, and mean IBI are the same in all cases, resulting in a constant OPI value of 0.83. However, whereas in examples A and C all blinks are complete, in B and D, 33% of blinks are incomplete. Similarly, blinks are very regular in A and B (as shown by a zero value of the coefficient of variation of IBI), but present a highly irregular pattern in C and D. Finally, when considering the percentage of ocular surface exposure time, the lowest and highest values correspond to A (regular and complete blinking pattern) and D (irregular and incomplete blinking pattern), respectively. It may be noticed that the percentage of ocular surface exposure times of B and C are very similar, although ocular surface exposure originates mainly from incomplete blinking in B and from irregular blinking in C. Given similar values of ocular surface exposure and a constant mean IBI of 12 s, examples B and C would result in a similar OPI 2.0 value, i.e., practitioners would lack precise information to assist them when advising these particular patients to improve their blinking habits.

### 2.2. Study Sample

To explore blinking parameters, a pilot study was designed in which 3 min video recording segments were captured while participants conducted three different tasks. Eight participants (4 females) with ages ranging from 29 to 70 years (mean ± standard deviation of 52.0 ± 16.6 years) were recruited. Participants were in good general and ocular health and had no known neurological disorders or took any medications that could exacerbate dry eye. All participants had binocular corrected distance and near visual acuity ≥1.0 (decimal).

Participants provided written informed consent after the nature of the study was explained to them, although they were not explicitly informed that blinking would be monitored until after completing the study. The study was conducted in accordance with the tenets of the Declaration of Helsinki of 1975 (as revised in Tokyo in 2004) and received the approval of an Institutional Review Board (Universitat Politècnica de Catalunya).

### 2.3. Baseline and Reading Conditions

Three different experimental configurations were tested (baseline and two reading conditions). During the baseline condition, subjects were instructed to observe in silence a high-contrast landscape picture pasted on the wall at 2 m and eye level. Reading conditions required participants to read a text on a computer display and in hard-copy format. In both reading conditions, we utilized a compilation of brief and accessible stories written by Quim Monzó, a Catalan author. The texts were displayed using consistent formatting, including the Arial typeface, font size of 9, line spacing of 1.15, and a similar word count per page.

Electronic reading took place on a panoramic 24 inch, 16:9 liquid crystal display (TFT-LCD) set to a resolution of 1920 per 1080 pixels, 32 bit color configuration, contrast ratio 700:1, and 75 Hz refresh rate. Text was presented at a 100% level of magnification. The level of luminance emitted by the computer display (210 cd/m^2^) was measured with a light meter GOSSEN MAVOLUX 5032 (GOSSEN Foto- und Lichtmesstechnik GmbH, Nürnberg, Germany) incorporating the luminance accessory and adjusted to allow comparison with the hard-copy text format (140 cd/m^2^). Hard-copy reading was conducted by pasting the text, printed in A4 size, over the switched-off computer display, thus ensuring the same relative distance and viewing angle in both reading configurations.

To ensure variability, we employed block randomization to assign each participant a unique order of experimental conditions. Both the baseline and reading sessions were conducted on the same day, specifically between the hours of 10:00 a.m. and 2:00 p.m. and for each participant all measurements were completed in about 18 min. Room temperature was maintained at about 20 °C (±2 °C). Background illumination was between 750–800 lx and was provided by diffuse lighting.

### 2.4. Video Recording and Analysis

Throughout the baseline and reading sessions, we utilized a Canon Legria HF M307 (Canon España S.A., Alcobendas, Madrid, Spain) to capture video footage of the participants’ eyes. This camera allowed for high-quality image capture at 3.3 MP with a resolution of 1920 × 1080 and a frame rate of 60 frames per second. For each experimental condition 3 min video recordings were obtained, although for analysis purposes the first two minutes of all video captures were discarded, that is, blinking was assessed during the last 1 min video segment of each recording.

A custom-made blink counting application was created to streamline the video analysis process. To assist with tracking time, a horizontal grating consisting of 60 small squares represented one minute of video recording, with each square symbolizing one second. By pressing predefined keyboard keys, the occurrence of complete or incomplete blinks could be marked (multiple blink events could be marked within each one-second square). As mentioned previously, we classified blinks as complete when the cornea was entirely obscured upon blink completion. In cases where the cornea remained partially visible, blinks were categorized as incomplete. We disregarded minor twitches or lid tremors. After reviewing each minute of video recording, the software provided data on the overall blink count and the percentage of incomplete blinks. The examiners assessed each one-minute segment of video recording while the software operated discreetly in the background. Each examiner independently assessed the blink parameters, and in the event of any discrepancies with their individual findings, a collaborative frame-by-frame analysis was conducted to examine specific blinking events. This joint analysis was only performed when necessary to reconcile differing results and ensure accurate assessment of blink parameters.

A typical 1 min segment blink assessment with the blink counting application is shown in Figure 2. As noted above, in real life conditions the first and last blinks do not always occur at the start and end of the 1 min segment. In these cases, a blink is assumed to occur immediately before and after the 1 min segment, that is, at the −1 and 61 s marks. This assumption, which is required when the analysis is limited to a single 1 min segment, may only result in an underestimation of ocular surface exposure.

In this example, blink parameters are as follows: SEBR = 8 blinks per minute; 25% of incomplete blinks; IBI coefficient of variation = 39%. Given a TFBUT of 7 s, the time percentage of ocular exposure when only complete blinks are considered is (7 + 9) seconds over 60 s, i.e., 26.67% of the time.

### 2.5. Tear Film Break-Up Time Assessment

Before the start of the experimental session, TFBUT was assessed by following the recommendations published in the Report of the International Dry Eye Workshop (DEWS) [20]. Three measurements of TFBUT were conducted and the average value was used. Participants were allowed ample time to recover after tear film break-up time assessment prior to blink evaluation.

### 2.6. Data Analysis

The aim of this study was to present a proof of concept of the relevance of assessing three different blink parameters and to illustrate this by conducting a short trial with a limited number of participants. Therefore, given the exploratory nature of this pilot study, data were subjected to neither descriptive nor inferential statistical analysis.

## 3. Results

Table 2 presents the values of age, gender, and TFBUT of the eight participants in the pilot study, together with the outcome of SEBR, percentage of incomplete blinks, blink regularity (coefficient of variation of IBI), and time percentage of ocular exposure (only complete blinks were considered) for each participant and experimental condition (baseline, electronic display, and hard-copy text). Overall, both reading conditions resulted in a reduction in SEBR and an associated loss of blink regularity (larger values of the coefficient of variation of IBI), although in some participants blink regularity was worse during baseline conditions. Interestingly, the percentage of incomplete blinks decreased during both reading conditions, with a noticeable trend towards more incomplete blinks during electronic display reading (given the significant reduction in SEBR during both reading conditions, however, these calculations may be interpreted with caution). Finally, a trend was also evidenced in which the time percentage of ocular surface exposure increased when participants were conducting a reading task, although in some of them, this trend was reversed.

As it may be expected, larger time percentages of ocular surface exposure were commonly associated with a reduction in SEBR, which leads to longer individual IBI values, with the worse outcomes of ocular exposure corresponding to those participants with the shortest TFBUT values. On the contrary, participants with the highest SEBR values usually displayed insignificant ocular surface exposure, irrespective of actual blink amplitude and regularity. However, ocular surface exposure was still found in some cases with high SEBR, given a wrong combination of TFBUT, blink amplitude, and regularity. Refer, for instance, to the baseline results of the 69 years old female, TFBUT = 4 s, SEBR = 15 blinks/minute, percentage of incomplete blinks = 46.0%, blink regularity = 55.9%, ocular exposure = 42.1%. In effect, this value of ocular exposure is comparable to that of the 45 years old female with TFBUT = 5 s, although in this participant’s case ocular exposure is mainly a consequence of reduced SEBR (4 blinks/minute) and high blink irregularity (130.5%), with no incomplete links registered in baseline conditions. For comparison purposes, it may also be interesting to review the results of both 44 years old males, with TFBUT = 10 s and relatively similar SEBR, blink amplitude, blink regularity, and ocular exposure in baseline conditions. A subsequent change during reading conditions in SEBR, blink amplitude, and regularity in one of these participants led to very different time percentages of ocular exposure.

## 4. Discussion

The results of the present pilot study confirmed that ocular surface exposure may occur in different degrees according to the particular combination of SEBR, blink amplitude, blink regularity, and TFBUT encountered on each patient or elicited by each experimental condition, thus giving support to the need to carefully assess all blink parameters to better advice patients regarding reeducation of their blinking habits. In effect, as indexes of ocular protection, neither the OPI nor the OPI 2.0 provide sufficient evidence to allow practitioners an informed decision, although both indexes have proven useful to assess tear film substitutes or to differentiate between normal and dry eye patients [3,4,7]. Similarly, the evaluation of SEBR and TFBUT alone may be inadequate to appraise the risk of ocular surface exposure, as it was evidenced in some of the participants of the pilot study in whom, albeit large values of SEBR, ocular exposure was still significant. In these participants, ocular exposure originated from a combination of incomplete blinking and increased irregularity, although the actual impact of these factors was not always balanced, with participants in whom irregularity had a superior influence while in others ocular exposure resulted mainly from incomplete blinking. Indeed, previous research under similar controlled experimental conditions revealed that the choice of reading support had a significant effect on blink rate and completeness but failed to consider blink regularity [17]. Aiming at a complete characterization of blinking parameters, the same experimental configuration was implemented.

In contrast, the OPI 2.0 is superior to the present approach, given that ocular surface exposure is monitored in real-time and measured as the actual area of surface exposure, rather than as a time percentage. However, this clear benefit is penalized by the need to conduct measurements in a controlled environment, with sodium fluorescein and a slit-lamp, which are not practical to implement in daily life conditions such as in an office or classroom. Indeed, the purpose of showcasing this proof of concept is to invite future researchers to work toward the development of real-time blink monitoring strategies. In this regard, several recent efforts have been conducted to detect blinking using image analysis and taking advantage of the ubiquity of cameras integrated into laptop computers and hand-held devices [21,22,23]. As far as we know, however, these attempts remain unsatisfactory, as they do not yet permit to distinguish between complete and incomplete blinks and are commonly not easy to implement in real-time, requiring posterior video image processing. One could envisage a near future, however, where technology would allow real-time automated video monitoring of all blinking parameters, with the possibility of giving feedback to users, aiming at reeducation.

It must be noted that the present approach is not free of assumptions requiring further verification. Thus, on the one hand, it is unclear how the area of ocular surface exposure increases over time after tear film break-up [6,11,24,25] and whether the type and area of ocular surface exposure is an intrinsic characteristic of each individual, or depends on a combination of several factors, such as but not limited to, individual traits (e.g., lid margin functionality and tear film composition), TFBUT, experimental conditions, prior ocular surface exposure events, etc. Therefore, it remains a subject of future research to determine the type (linear, exponential, etc.) and the factors governing the relationship between area and time of ocular surface exposure.

Related to the previous consideration, the present approach assumes that TFBUT remains stable over time, that is, that TFBUT measurements are repeatable at each IBI. Previous research has noted modest repeatability and reproducibility of TFBUT measurements [26,27,28]. However, these authors measured TFBUT in a controlled setting, instructing patients to perform several complete blinks between each measurement, thus restoring the integrity of the tear film to the original conditions. It is unclear whether baseline TFBUT measurements provide valid information on real-time tear film stability while participants are conducting the reading tasks.

Similarly, it is possible that, even with incomplete blinks, partial tear film renewal occurs over the area of previous corneal exposure, that is, not all incomplete blinks may be considered as a no-blink condition. It may be interesting to explore whether a previous long exposure time may influence TFBUT after the restoring blink, particularly if the intervening blink is not complete. Indeed, an alternative approach to determine SEBR and blink regularity could be implemented in which only complete blinks are considered. Given the assumed inadequacy of an incomplete blink to fully restore tear integrity, this alternative could provide a more realistic picture of the actual affectation to the ocular surface. Notwithstanding these considerations, however, published literature on blink parameters commonly expresses SEBR as blinks per minute, without determining whether each blink is complete or incomplete.

It must be noted that, as a pilot study, this research lacks the required power to conduct a complete statistical analysis, and results are summarized in descriptive terms alone. However, given the individual differences observed in this reduced sample of patients, it may be assumed that a larger study would reveal a noticeable contribution of blink regularity to ocular surface exposure. Further research is required to confirm this suggestion.

In conclusion, the present study provides a proof of concept to highlight the need to explore other blink parameters, including complete blinks and blink regularity, in addition to the customary SEBR, as well as TFBUT, to better appraise the risk of ocular surface exposure. Furthermore, given the documented high sensitivity of these blink parameters to changes in measuring conditions, it would be recommended to assess them while patients conduct real-life activities such as reading or using visual display terminals. In view of the large percentage of users reporting ocular dryness during and after prolonged computer work, these considerations may be of particular relevance when providing advice to these users aiming at a reeducation of their blinking habits.

## Figures and Tables

**Figure 1 diagnostics-13-02362-f001:**
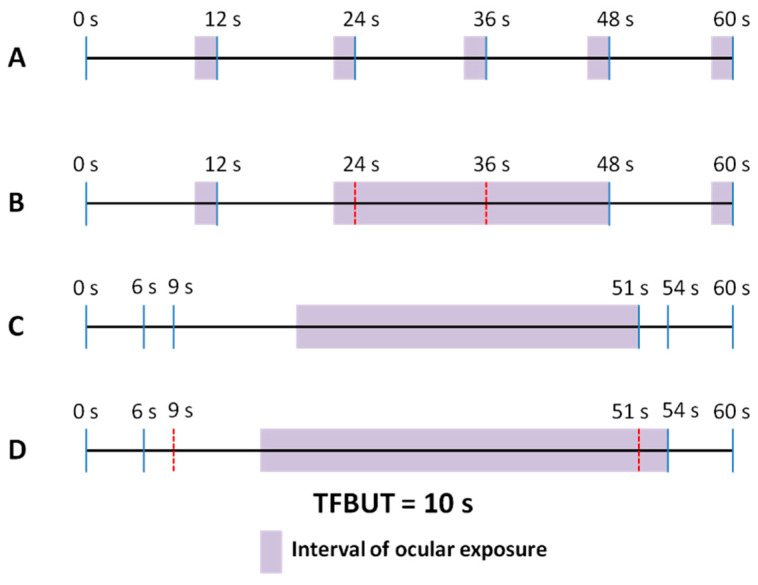
Four different blink patterns with the same mean interblink interval of 12 s over 1 min assessment. (**A**). Complete blinks in a regular pattern; (**B**). Complete and incomplete blinks in a regular pattern; (**C**). Complete blinks in an irregular pattern; (**D**). Complete and incomplete blinks in an irregular pattern. (Complete blinks are shown as continuous vertical blue line segments, incomplete blinks as discontinuous vertical red line segments). Tear film break-up time (TFBUT) is 10 s. Shadowed areas represent instances of ocular exposure.

**Figure 2 diagnostics-13-02362-f002:**
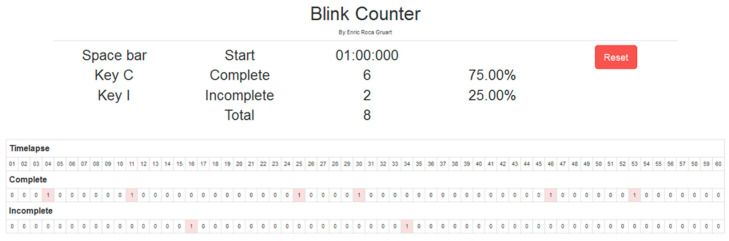
Screen capture of the blink counter application used to record complete and incomplete blinks over 1 min video recording segments.

**Table 1 diagnostics-13-02362-t001:** Blink and tear film parameters of the four examples presented in Figure 1.

	TFBUT (s)	SEBR (Blinks/min)	<IBI> (s)	OPI	Incomplete Blinks (%)	Regularity: Coefficient of Variation of IBI (%)	Percentage Exposure Time (%)
A	10	6	12	0.83	0	0	16.7
B	10	6	12	0.83	33	0	50
C	10	6	12	0.83	0	140.3	53.3
D	10	6	12	0.83	33	140.3	63.3

TFBUT: Tear film break-up time; SEBR: Spontaneous eyeblink rate; IBI: Interblink interval; <IBI>: Mean interblink interval; OPI: Ocular Protection Index.

**Table 2 diagnostics-13-02362-t002:** Age, gender (Male/Female), and tear film break-up time (TFBUT) of all participants. Spontaneous eyeblink rate (SEBR), percentage of incomplete blinks, blink regularity (coefficient of variation of IBI), and time percentage of ocular surface exposure (when only complete blinks are considered) are shown for each participant and experimental condition (baseline, electronic display, and hard-copy text).

Participants			Baseline Conditions				Electronic Display Reading				Hard-Copy Text Reading			
Age (y)	Gender (M/F)	TFBUT (s)	SEBR (Blink/min)	Incomplete Blinks (%)	Blink Reg. (%)	Ocular Exposure (%)	SEBR (Blink/min)	Incomplete Blinks (%)	Blink Reg. (%)	Ocular Exposure (%)	SEBR (Blink/min)	Incomplete Blinks (%)	Blink Reg. (%)	Ocular Exposure (%)
44	M	10	19	39.3	48.7	8.0	5	0	62.6	38.9	3	0	93.4	34.1
29	F	12	31	21.1	46.1	0.0	18	15.3	57.3	0.0	16	29.0	57.2	1.8
70	M	6	28	68.6	56.4	6.0	5	20	89.9	66.7	2	26.0	108.4	75.5
69	F	4	15	46.0	55.9	42.1	5	26.0	77.7	70.0	5	26.0	48.2	66.7
44	M	10	22	51.9	45.5	6.8	18	36.0	27.7	0.0	21	6.9	33.7	0.0
45	F	5	4	0	130.5	56.2	21	13.8	46.0	6.8	18	23.0	101.9	7.0
41	M	5	32	38.0	67.5	13.0	3	34.3	80.5	59.2	4	0	84.7	83.3
74	F	4	24	41.3	54.1	8.8	16	6.9	49.2	12.1	18	0	44.9	15.8

## Data Availability

Data may be available upon reasonable request to the authors.

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
