# Peer review of "Loss of Blink Regularity and Its Impact on Ocular Surface Exposure"

_diagnostics, 2023, doi:10.3390/diagnostics13142362_

Round 1
Reviewer 1 Report
Thank you very much for allowing me to review the manuscript entitled “Loss of Blink Regularity and Its Impact on Ocular Surface Exposure”.
Considering the demerits of conventional blink-related parameters, authors suggested new parameters such as blink regularity and time percentage of ocular exposure in this manuscript. I think that it is very good idea. However, the clinical usefulness of these new parameters was not proved enough. At first, please increase the number of subjects and perform statistical analysis. I am looking forward to reading authors’ next report.
Author Response
Thank you very much for allowing me to review the manuscript entitled “Loss of Blink Regularity and Its Impact on Ocular Surface Exposure”.
Considering the demerits of conventional blink-related parameters, authors suggested new parameters such as blink regularity and time percentage of ocular exposure in this manuscript. I think that it is very good idea. However, the clinical usefulness of these new parameters was not proved enough. At first, please increase the number of subjects and perform statistical analysis. I am looking forward to reading authors’ next report.
RESPONSE: Thank you for these kind words regarding our manuscript. The idea behind this article was to present a proof of concept, a description of a novel approach towards full blink characterization, including blink regularity, which has been largely neglected by the scientific community until now. To support our idea, we conducted a pilot study with a limited number of subjects but under well-controlled settings, and summarized our findings in a narrative-descriptive manner as a complete statistical analysis was not possible at this stage. We aim to follow-up this study with a more extensive research enrolling a larger group of participants. However, we also feel that by offering the scientific community this initial proof of concept, other researchers may start working on this.
We have added this paragraph to the discussion of the manuscript, to reflect the limitations of our approach: “It must be noted that, as a pilot study, this research lacks the required power to conduct a complete statistical analysis, and results are summarized in descriptive terms alone. However, given the individual differences observed in this reduced sample of patients, it may be assumed that a larger study would reveal a noticeable contribution of blink regularity to ocular surface exposure. Further research is required to confirm this suggestion.”
Reviewer 2 Report
This is an article entitled “Loss of blink regularity and its impact on ocular surface exposure (diagnostics-2451457)”.
The idea is good but the English needs revision.
Abstract
- Confusing. Please rewrite.
Introduction
- OPI = TFBUT / <IBI> è what should the units of the data be? Please admit.
Materials and Methods
- Ok.
Results
- Ok.
Discussion
- Ok.
Tables
- Ok.
References
- Ok.
This is an article entitled “Loss of blink regularity and its impact on ocular surface exposure (diagnostics-2451457)”.
The idea is good but the English needs revision.
Abstract
- Confusing. Please rewrite.
Introduction
- OPI = TFBUT / <IBI> è what should the units of the data be? Please admit.
Materials and Methods
- Ok.
Results
- Ok.
Discussion
- Ok.
Tables
- Ok.
References
- Ok.
Author Response
This is an article entitled “Loss of blink regularity and its impact on ocular surface exposure (diagnostics-2451457)”.
The idea is good but the English needs revision.
RESPONSE: Thank you. We have revised and improved the structure of some sentences of the manuscript.
Abstract
- Confusing. Please rewrite.
RESPONSE: We have revised the abstract to improve its clarity.
Introduction
- OPI = TFBUT / <IBI> è what should the units of the data be? Please admit.
RESPONSE: Thank you. As noted in line 34, the OPI values are dimensionless. OPI is a ratio between TFBUT (in seconds) and mean IBI (also in seconds). Therefore, it lacks units.
Reviewer 3 Report
Interesting study, well done. Just one question: Why did you choose one group reading hard copies and one group looking at electronic displays? Was there a difference? Are there studies looking also at the difference of hard copies and electronic displays? Please add a short comment in the discussion about this aspect.
Author Response
Interesting study, well done. Just one question: Why did you choose one group reading hard copies and one group looking at electronic displays? Was there a difference? Are there studies looking also at the difference of hard copies and electronic displays? Please add a short comment in the discussion about this aspect.
RESPONSE: Thank you for your king appraisal. Regarding your question, yes, this study was part of a larger effort (mainly published here: https://doi.org/10.1167/iovs.15-16967) in which we examined blink rate and blink completeness in several reading conditions, including hard-copy and electronic displays. We uncovered a significant influence of the type of reading support on blink parameters. Therefore, we considered that similar experimental settings would be adequate to explore changes in blink regularity. We have added this sentence to the discussion of the manuscript: “Indeed, previous research under similar controlled experimental conditions revealed that the choice of reading support had a significant effect on blink rate and complete-ness, but failed to consider blink regularity [17]. Aiming at a complete characterization of blinking parameters, the same experimental configuration was implemented.”
Round 2
Reviewer 1 Report
The authors only proposed new parameters which clinical usefulness was not proved sufficiently. Again, authors should increase the number of subjects and perform statistical analysis.